# Masking Behaviors in the Absence of Local Mandate—An Observational Study from Hillsborough County, Florida

**DOI:** 10.3390/ijerph192315982

**Published:** 2022-11-30

**Authors:** Jessica Pecoraro, Chighaf Bakour, Alison Oberne, Amber Mehmood

**Affiliations:** College of Public Health, University of South Florida, Tampa, FL 33612, USA

**Keywords:** COVID-19, prevention, mask, social distancing, masking, policy, mandates, communication, disparities

## Abstract

The purpose of this study was to understand the predictors of masking—especially age, race/ethnicity and gender—in Hillsborough County Florida, a region without mask mandates. Masking and social distancing behaviors of individuals were observed in Hillsborough County during one-week intervals in July 2021, August 2021 and Late September—early October 2021. Demographic and behavioral observations were recorded and logistic regression was utilized to determine the odds ratio of wearing a mask amongst various groups. Overall, masking ranged from 36.80% to 48.64%, peaking during the second observation period. Masking rates were highest amongst people of color, women and seniors. Establishments posting mask-negative language, such as “masks NOT required,” saw a 46% decrease in the odds of masking compared to establishments without mask-related signage (thereby defaulting to no mandate). Understanding who engaged in masking and social distancing behaviors will provide local public health officials with a deeper understanding of the effectiveness of previously used strategies, which can be leveraged in future surges of COVID-19 and other emergencies to create maximum impact. Lessons learned regarding policy implementation and understanding patterns of uptake of health guidelines are important for the continuous improvement of public health practice.

## 1. Introduction

Since its emergence in 2020, SARS-CoV-2 has cost over one million lives in the United States alone [1]. The rapid spread of this novel virus necessitated an immediate and rapid introduction of public health prevention and mitigation strategies, including masking and social distancing. As our understanding of the virus, its transmission, reproduction number (R0) and variants evolved, so did the guidelines or strategies that were encouraged by health officials and the subsequent uptake of those strategies [2]. In the face of crisis, people responded in a variety of ways, from adhering to the words of experts to rejecting all advice and public health guidance [3]. These individual pandemic responses varied across regions and amongst different populations [3].

In the early stages of the pandemic, prior to vaccines becoming readily available, masking, social distancing and handwashing formed the bedrock of the public health response [4]. These strategies require widespread adherence to create meaningful community-level transmission reduction. Use of these public health strategies varied around the globe as each country assessed its own risk and capacity to take specific prevention steps [5,6]. Some countries enacted full lockdowns to enforce social distancing, whereas others left it up to their citizens to decide their level of comfortable interaction [7]. Masking was strongly recommended as a preventive measure, but countries and states varied in terms of mandating masking in public places, leading to different outcomes, considering that masking must be implemented with high fidelity to maximize effectiveness [8,9]. An early evaluation of COVID-19 state-level policies showed greater declines in daily COVID-19 cases after issuing mask mandates compared with states that did not have mandates in place [10]. Despite this evidence, there was no U.S. federal mask mandate (though masking was recommended), leaving prevention guidelines to each state [11]. In states that opted to not enforce mask mandates, each locality implemented public health strategies that local leaders felt appropriate—leaving some neighboring localities with differing approaches. In places without state mandates, messaging varied even between local establishments, leading to an increased burden for local officials and confusion among residents.

Health disparities and inequities in access to care and education have been problematic in the U.S. and this global health emergency has further highlighted these gaps [12]. This is evident within the COVID-19 morbidity and mortality statistics, highlighting that certain populations have been impacted disproportionately [13]. Although it might be expected that different levels of transmission and severe illness would be seen based on the level of preventative steps being taken within the community, disparate rates can be seen between different demographic groups within the same community. Older individuals, for instance, were at an increased risk of hospitalization and/or death from COVID-19 than were younger individuals [1,14]. Black and Hispanic individuals also experienced comparatively poor COVID-19 outcomes, relative to whites [1,13,14].

Just as each state and sometimes each locality, approached the pandemic differently, each region had its own social and economic dynamics further compounding the variations in masking adherence. Although national guidance encouraged masking, the Florida governor indicated that establishments could not require masking [15,16]. This resulted in a wide array of advertisements among establishments, ranging from signs reinforcing the fact that masks were “not required” to signs encouraging or even stating that masks were required and some establishments opted to take no advertised stance.

The pandemic created an immediate, urgent need to roll out strategies to minimize the transmission and societal health impacts of this novel virus, such as masking and social distancing. The current observational study was nested within the nationwide Systematic Observation of Mask Adherence and Distancing (SOMAD) study [17]. The SOMAD study analyzed patterns across the country, whereas this paper focuses exclusively on the patterns of masking behavior in Hillsborough County, Florida. The state of Florida did not mandate masking, allowing individual businesses, organizations and institutions under federal control to decide on their own course of action, until ultimately banning businesses from requiring masks on 3 May 2021 [15].

With this background, the purpose of this study was to understand the predictors of masking in this region—especially age, race/ethnicity and gender; special attention was given to consider the impacts of policy and signage on mask wearing behaviors in Hillsborough County, Florida. Examining who engaged in masking and social distancing behaviors will provide local public health officials with a deeper understanding of the effectiveness of previously used strategies, which can be leveraged in future surges of COVID-19 and other public health emergencies to create maximum impact.

## 2. Materials and Methods

### 2.1. Population, Setting and Data Collection

Observations were conducted over three one-week time periods during July 2021 (Observation Period 1), August 2021 (Observation Period 2) and late September through early October 2021 (Observation Period 3) in Hillsborough County, Florida. Hillsborough County had a population of approximately 1,478,194 in 2021 that was predominantly White (73.3%), 50.8% female and had a median household income of $60,566 annually [18]. Subjects were observed at local businesses, such as grocery stores and malls in various parts of Hillsborough County and a large, urban university campus. Observers recorded data using a survey created in Qualtrics™ online survey tool (manufactured by Qualtrics in Provo, UT, USA), which allows for the creation of surveys, collection of responses and export of detailed responses for analysis. The survey included questions about the location, apparent demographics and COVID-19 prevention behaviors (i.e., masking and social distancing). All data were collected by observation and no contact was made with any subject. The timing of each entry is recorded automatically in Qualtrics. To ensure consistent, accurate observations, data collectors were trained in how to reliably estimate age and gender and how to use Qualtrics. In total, there were eight observers/data collectors paired in four groups, with one of the two observers at each location entering the survey responses for each observed individual. Data were collected across all days of the week and at differing times of day to account for possible variability in masking and social distancing adherence. Each pair of data collectors went to a specific location based on a pre-set schedule, to avoid any overlap in data collection. Demographic and behavioral characteristics were noted during each of the three 1-week observation windows. IRB approval was obtained for this study.

### 2.2. Variables

Masking behavior was observed and categorized as “Full Mask” when individuals wore a mask over their nose and mouth, “Partial Mask” when individuals wore a mask covering their nose or mouth but not both, “Visible Mask NOT on” when individuals were observed to be in possession of a mask, but it did not cover their nose and/or mouth and “No Mask” when no mask was visible. For the purposes of regression analysis, the masking variable was then dichotomized into two Y/N variables, one for full masking that covers the nose and mouth vs. other (partial masking, no mask, etc.) and the second for any masking (full or partial) vs. no mask. Approximations were made for age such that ages 3–11 were categorized as “Child,” ages 12–17 were categorized as “Teen,” ages 18–64 were categorized as “Adult” and ages 65+ were categorized as “Senior.” The reference group for age was the “Adult” group because it included the most observations. Data for 0–2-year-olds were recorded but not included in the analysis, as masks were not recommended by the CDC or the American Academy of Pediatrics for children 2 years of age and younger [19]. Race/ethnicity was observed and categorized as Asian, Black/African American, Hispanic, White and Other. For analysis purposes, the Asian and Other groups were combined due to a small number of observations in those groups. The reference group for race/ethnicity was “White” as it was the largest of the race/ethnicity groups. Signage was recorded as “None,” “Masks Not Required,” “Masks Recommended” and “Masks Required;” however, for analysis purposes the “Masks Recommended” and “Masks Required” were combined into “Mask-positive language” due to a small number of “Masks Required” signs. “Masks Not Required” was termed “Mask-negative language.” No signage (“None”) was the reference group for signage.

### 2.3. Statistical Analysis

Descriptive statistics were calculated with frequencies and percentages. The “Mask Y/N” variables were utilized to conduct logistic regression analysis assessing the association between each predictor and masking behavior using each of the two masking variables separately. Individual unadjusted models were run for each of the following predictors: Observation period, age group, sex, race/ethnicity, location, weekday/weekend, physical distancing and signage, and each masking outcome variable. The fully adjusted models included all predictors. To examine potential effect modification by age and race/ethnicity the fully adjusted model was stratified by the age category and by race/ethnicity. All statistical analyses were performed using SAS^®^ 9.4 software, a statistical program manufactured by SAS Institute Inc. in Cary, NC, USA.

## 3. Results

A total of 6035 observations were recorded during the three observation windows: 1970 observations were recorded in the first observation period, while 2170 observations were recorded in the second period 2 and an additional 1886 observations were recorded in the third period 3. The overall prevalence of “full” masking was 36.8% in observation period 1, 48.64% in observation period 2 and 46.95% in observation period 3, while the prevalence of “any” masking (full or partial) over these same observation periods were 44.54%, 55.92% and 53.10%, respectively. As shown in Table 1, analysis of the demographic characteristics of the observed population by masking behavior revealed that full masking was most prevalent among seniors (57.22%), while among the age groups recommended to mask, teens were the least likely to be fully masked (31.69%). Full masking was also more prevalent among Asians (57.14%) and least prevalent among Whites (39.33%). Additionally, full masking was more likely in establishments that had signage stating masks were required (58.90%) and least likely in establishments with signage stating that masks were NOT required (22.15%).

During the first observation period 31.1% of children were observed fully masked. This behavior increased to 41.0% in the second observation period before decreasing to 36.3% in the third observation period. Figure 1 depicts the percentage of children fully masking over the three observation periods as compared to the aggregate age category (teens through seniors). These data highlight the increase in masking amongst both groups from the first to the second observation period followed by a decrease amongst children, while the older populations experienced a continued increase, albeit slight.

Overall masking behavior differences by race/ethnicity are illustrated in Figure 2. The highest prevalence of full masking (55.6%) was in racial groups that were neither White, Black, nor Hispanic, while the lowest (39.3%) was among Whites. The highest prevalence of partial masking was among Blacks (48.2%), while the lowest was among Whites (6.9%). When full masking was analyzed across the three observation periods, all race/ethnicity groups increased the rate of full masking from the first to the second observation period. A divergent pattern was seen from the second to the third observation period, where White, Black and Hispanic groups decreased full-masking rates, while the Other group further increased the rate of full masking.

Figure 3 highlights the differences in masking behaviors seen at the various establishments based on signage in a particular location. The results indicate that, as compared to establishments with no mask-related signage, individuals in establishments with “Mask-positive” signage had an increase in the odds of mask wearing. Conversely, individuals in locations with “Mask-negative” signage had a decrease in the odds of being fully masked as compared to those in establishments with no mask-related signage.

Unadjusted and adjusted odds ratios and 95% confidence intervals for the key predictor variables of interest are shown in Table 2. The fully adjusted model revealed a 58.8% increase in the odds of full masking during observation period 2 and a 52.7% increase during observation period 3, as compared to observation period 1. The model also revealed a statistically significant increased odds of full masking amongst Asians, Black/African Americans and “other” racial groups, as compared to the White population. Seniors were significantly more likely to fully mask compared with adults (OR = 1.71; 95% CI 1.47, 2.00) and females were 49% more likely to fully mask compared with males and 56% more likely than males to wear any mask (full or partial). Other significant predictors included increased odds of masking on weekends, compared with weekdays, presence of positive signage, Publix compared with Walmart, and full masking was more likely among those who observed physical distancing.

When the model was stratified by age, children had a 41.6% increase in the odds of full masking in observation period 2 and a 23.3% increase in observation period 3, as compared to observation period 1 (Table 3) During these same times, a combined age category including teens through seniors experienced a 56.1% increase and a 70.4% increase in the odds of masking in the second and third observation periods respectively, as compared to observation period 1 (Table 3). When the model was stratified by race/ethnicity, White and Black/African Americans both experienced a statistically significant increase in masking during the second observation period and a slightly lesser increase in masking in observation period 3, as compared to the first observation period (Table 4).

## 4. Discussion

This study provided an opportunity to capture some of the patterns of masking behaviors in Hillsborough County, Florida, where people of color, women and seniors were more likely to adhere to full masking, regardless of day and time, social distancing behavior, or observation period. As guidelines and policies were rolled out for SARS-CoV-2, they were met with a variety of reactions. Some people followed every guideline, while others rejected the guidance aimed at protecting the community, taking more of an “each man for himself” approach [2]. Our study not only demonstrated that in the absence of a strict mask mandate, individuals acted upon their own interpretation of the risk of disease susceptibility but also adhered to local guidelines wherever possible.

The presence of mask-positive signage at businesses was associated with increased odds of mask adherence despite the absence of a statewide mask mandate. The presence of signage with negative language (masks “not required”) was associated with decreased mask adherence compared to locations with no signage. Overall mask adherence ranged between 36.80% and 48.64%, peaking during observation period 2 (the last week in August 2021). Our study also captured a period that included the Delta wave surge, as well as the opening of in-person school sessions in Hillsborough County, FL. Both of these events were central to local masking decisions and are important to consider as a backdrop to the results of this study.

In Hillsborough County, as seen in other studies and other locations, Black/African American individuals were significantly more likely to mask than White individuals. Interestingly, all race/ethnic groups experienced an increase in masking from July to August, corresponding to the surge period of the Delta variant, which was again followed by a general decline in the prevalence of masking as the surge declined. Interestingly, the Other race category saw a continued increase in masking from the second to the third observation period. Understanding the driving factors of this divergent trend could help illuminate the nuances of effective motivators related to the highly dynamic issue of adherence to public health guidance.

Therefore, it is evident that despite “individual” behaviors, our study reconfirmed the findings from other nationwide studies that women, minorities and elderly’s adherence to facial covering is stronger and henceforth their risk perception or health beliefs may be different from other demographic groups in Florida, which may be a target for future health promotion endeavors [4,8]. In these studies, self-perceived risk of COVID-19, fear of the virus, perceived utility of masks, political orientation and behavior change in response to the pandemic were cited as important individual factors. Additionally, studies have noted that, when mandates are not in place, locally communicated policies may have an impact on behavior [20]. Even though, we did not study all of these variables, our study pointed out that certain demographic groups may not be averse to adopting mask mandates. This observation has important implications for effective risk communication by framing the message that appeals to a community’s health beliefs and collective risk perception, especially among vulnerable populations such as the elderly and school going children. Examples of such messaging include an integrated approach of public health awareness through social media and messages targeting positive public health behavioral change [21]. Together, these factors influenced preventative actions toward the pandemic, especially considering variations in timing, length and effectiveness of mask mandates with ongoing changes in SARS-CoV-2 transmission and the incidence of symptomatic disease [3,20,21,22,23].

Mask wearing among children attending in-person instruction was a topic heavily discussed and contested in Hillsborough County [24]. Throughout the pandemic, the American Academy of Pediatrics and CDC recommended mask wearing for children over 2 years of age; however, in Florida, the governor forbade establishments from requiring masks. In light of concerns surrounding the Delta variant, on 21 August 2021, Hillsborough County Public Schools decreed that masks would be required in schools, even at the peril of losing state funding [24]. This extra level of caution mirrored the seemingly increased levels of concern within the public, as child mask-wearing increased significantly in late August (the second observation period). Due to increasing pressure from state government, bans against requiring masks and public debate over the necessity of children wearing masks, this policy was rescinded in October, making masks optional, despite continued CDC guidance recommending masks. In late September and early October (observation period 3), the odds of masking amongst children dropped from the peak seen in August but remained higher than what was observed in July. Since the incidence and morbidity of SARS-CoV-2 in children were lower as compared to adults, policy makers overlooked the evidence in favor of children wearing face coverings, making policy decisions that opened an avenue for increased community spread of SARS-CoV-2. Studies in other states highlighted the importance of facial coverings in school-going children, especially indoors and with extended periods of in-person exposure [25]. Enacting policies that conflict with the guidance of public health officials added confusion to an already stressful situation, thus making it difficult for individuals to know who to believe or the best course of action to take. Ensuring that the public understands the critical nature of all components of COVID-19 prevention strategies may help bolster adherence in the future, including adherence amongst seemingly less at-risk populations such as, in this case, children.

As discussed earlier, local signage was found to have an impact on people’s behavior, highlighting the critical importance of local messaging when rolling out public health policies, guidance and initiatives. Locations with signs recommending/requiring masks had higher masking rates despite no official state mandate. Locations that reiterated that masks were “not required” saw lower masking rates than locations that did not have any mask-related signage, thus, defaulting to the same state policy. While this may have partially been due to some people choosing to only enter establishments requiring masks, it is likely that the posted “not required” stance of a local establishment influenced the behaviors of those patrons who might have opted to wear a mask when no “negative” messaging was present. It has been previously noted in tighter (vs. looser) states, that mask-wearing was considered a civic duty, whereas, in southern U.S. states, masking was seen as spoiling one’s public image [23]. This cultural preference was highlighted in our study and was more pronounced for white adult men than any other ethnic, racial, gender, or age denomination. Understanding local cultural preferences and the influence of local messaging can help shape future approaches to public health initiatives.

This study has several strengths including that it was conducted over several time periods to gain an understanding of masking in Hillsborough County, Florida. Observations were made at a variety of locations, at various days and times, to capture a broad snapshot of the local population’s behaviors. This data was also collected during the Delta wave so differences in masking could be observed as rates of COVID-19 increased within the population. Finally, the use of a standardized tool and training for data collection helped to minimize bias in our results. Limitations in this study include the possibility that the subjects observed may not be representative of the entire population of Florida, despite efforts to capture a broad swath of the local population by including different neighborhoods, businesses and diverse locations. There is a potential for incidental repeated observations while collecting data at a given location. Age, sex, race and ethnicity were the best estimates, especially in the subset of masked population. Therefore, despite training, errors in estimated age category and race may persist and is a limitation to keep in mind when considering the results.

## 5. Conclusions

Lessons learned regarding policy implementation and understanding patterns of uptake of health guidelines are important for the continuous improvement of public health practice. This study highlights the differences in masking between different demographic groups, as well as differences in masking that occur in the presence of different types of mask-related signage. In the case of public health measures that require high fidelity of implementation to reach maximum effectiveness, it is important to use these lessons learned to devise risk communication strategies in the future. With SARS-CoV-2 constantly producing new immune-evasive variants, it is likely to continue to be a major public health issue for years to come. Understanding the influence of national guidance, state mandates, public health messaging and local policies on community-level behavior is important in effectively implementing policies and initiatives for future health crises caused by SARS-CoV-2 or by other pathogens. Additional research should continue to explore the level of influence different sources of guidance have on health belief and risk perception driving positive behavior change.

## Figures and Tables

**Figure 1 ijerph-19-15982-f001:**
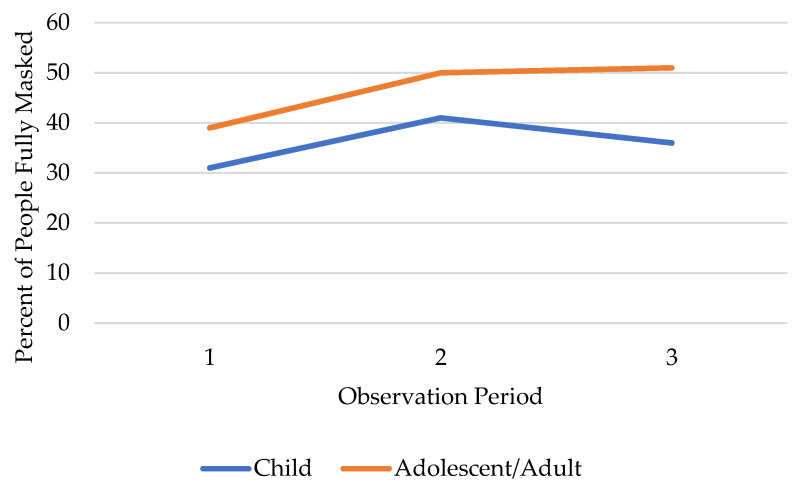
Masking trends over time in children and adolescents/adults.

**Figure 2 ijerph-19-15982-f002:**
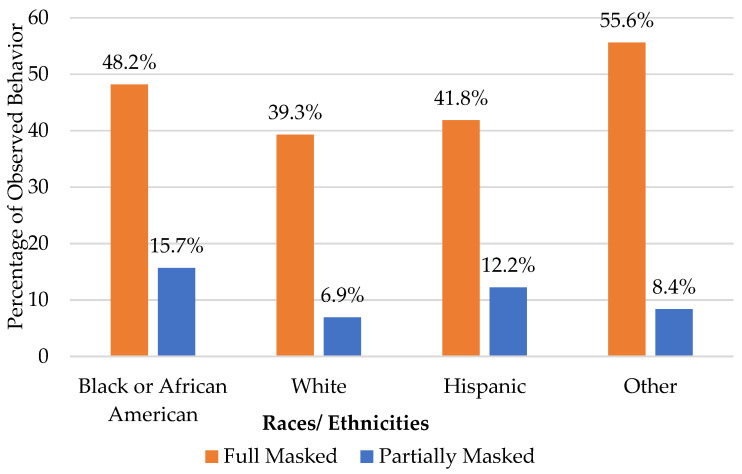
Overall masking behaviors by race/ethnicity.

**Figure 3 ijerph-19-15982-f003:**
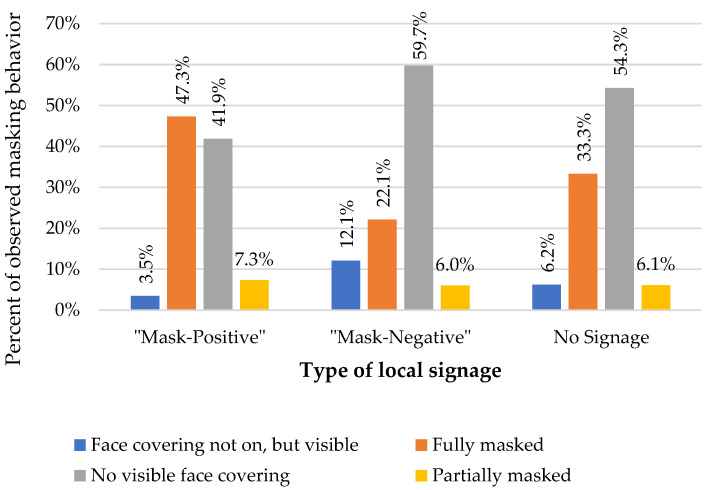
Masking behavior observed by local signage.

**Table 1 ijerph-19-15982-t001:** Overall masking behavior by variable.

	Full Masking (%)	Partial Masking (%)	Visible Mask NOT on (%)	No Mask (%)
Race/Ethnicity				
Asian	224 (57.14)	10 (2.55)	20 (5.10)	138 (35.20)
Black/AA	1026 (48.21)	224 (10.53)	110 (5.17)	768 (36.09)
Hispanic	544 (41.85)	116 (8.92)	43 (3.31)	597 (45.92)
Other	28 (45.90)	6 (9.82)	2 (3.28)	25 (40.92)
White	846 (39.33)	71 (3.30)	78 (3.63)	1156 (53.74)
Age Group				
Infant/Toddler	34 (15.04)	7 (3.10)	2 (0.88)	183 (80.97)
Child	253 (36.09)	76 (10.84)	44 (6.28)	328 (46.79)
Teen	167 (31.69)	45 (8.54)	28 (5.31)	287 (54.46)
Adult	1711 (46.27)	242 (6.54)	145 (3.92)	1600 (43.27)
Senior	503 (57.22)	57 (6.48)	34 (3.87)	285 (32.42)
Sex				
Female	1593 (48.19)	250 (7.56)	139 (4.20)	1324 (40.05)
Male	1072 (39.53)	173 (6.38)	113 (4.17)	1354 (49.93)
Other	3 (20.00)	4 (26.67)	1 (6.67)	7 (46.67)
Signage				
Not Required	33 (22.15)	9 (6.04)	18 (12.08)	89 (59.73)
No Sign	355 (33.46)	65 (6.13)	66 (6.22)	575 (54.19)
Recommended	2237 (47.10)	349 (7.35)	165 (3.47)	1998 (42.07)
Required	43 (58.90)	4 (5.48)	4 (5.48)	22 (30.14)
Weekday/Weekend				
Weekday	1422 (42.51)	235 (7.03)	132 (3.95)	1556 (46.52)
Weekend	1246 (46.37)	192 (7.15)	121 (4.50)	1128 (41.98)

**Table 2 ijerph-19-15982-t002:** Masking by observation period—full mask and any mask (unadjusted and adjusted).

Variable	Full Mask Overall OR (95% CI)	Any Mask Overall OR (95% CI)
Unadjusted Models		
Observation Period		
1	Ref.	Ref.
2	1.63 (1.44, 1.85)	1.59 (1.40, 1.79)
3	1.52 (1.33, 1.72)	1.41 (1.24, 1.60)
Adjusted Models		
Observation period		
1	Ref.	Ref.
2	1.59 (1.39, 1.81)	1.60 (1.40, 1.82)
3	1.53 (1.34, 1.76)	1.46 (1.27, 1.68)
Race/Ethnicity		
Black/AA	1.70 (1.48, 1.94)	2.28 (1.99, 2.61)
Hispanic	1.32 (1.13, 1.53)	1.64 (1.41, 1.90)
White	Ref.	Ref.
Other	2.27 (1.82, 2.83)	2.32 (1.85, 2.90)
Age Group		
Child	0.69 (0.47, 0.71)	0.81 (0.68, 0.96)
Teen	0.57 (0.47, 0.71)	0.67 (0.55, 0.81)
Adult	Ref.	Ref.
Senior	1.71 (1.47, 2.00)	1.80 (1.54, 2.12)
Signage		
No Signage	Ref.	Ref.
Mask-Negative Language	0.51 (0.33, 0.78)	0.54 (0.36, 0.81)
Mask-Positive Language	1.03 (0.79, 1.33)	1.09 (0.85, 1.40)
Distancing		
No	Ref.	Ref.
Yes	1.22 (1.08, 1.37)	1.17 (1.04, 1.32)
Location Type		
Walmart	Ref.	Ref.
Target	1.24 (1.004, 1.52)	1.09 (0.89, 1.35)
Publix grocery Shop	1.31 (1.10, 1.56)	1.16 (0.97, 1.39)
Campus Building	0.77 (0.60, 0.99)	0.72 (0.57, 0.92)
Mall	0.65 (0.51, 0.83)	0.64 (0.51, 0.81)
Weekday/Weekend		
Weekday	Ref.	Ref.
Weekend	1.14 (1.02, 1.28)	1.16 (1.04, 1.30)
Gender		
Male	Ref.	Ref.
Female	1.49 (1.34, 1.66)	1.56 (1.40, 1.74)

**Table 3 ijerph-19-15982-t003:** Adjusted Odds ratios for full masking, stratified by age group.

Variable	Child OR (95% CI)	Adolescents–Seniors OR (95% CI)
Obs period		
1	Ref.	Ref.
2	1.42 (0.95, 2.12)	1.56 (1.36, 1.79)
3	1.23 (0.82, 1.85)	1.70 (1.47, 1.97)
Race/Ethnicity		
Black/AA	1.46 (0.95, 2.24)	1.64 (1.43, 1.87)
Hispanic	1.39 (0.87, 2.19)	1.21 (1.03, 1.41)
White	Ref.	Ref.
Other	4.02 (1.81, 8.93)	1.85 (1.48, 2.32)
Signage		
No Signage	Ref.	Ref.
Mask-Negative Language	0.30 (0.09, 0.97)	0.49 (0.31, 0.78)
Mask-Positive Language	1.19 (0.78, 1.81)	1.78 (1.52, 2.08)
Distancing		
No	Ref.	Ref.
Yes	1.42 (0.93, 2.16)	1.22 (1.08, 1.37)
Weekday/Weekend		
Weekday	Ref.	Ref.
Weekend	1.60 (1.15, 2.22)	1.15 (1.02, 1.29)
Gender		
Male	Ref.	Ref.
Female	0.68 (0.50, 0.94)	1.60 (1.43, 1.80)

**Table 4 ijerph-19-15982-t004:** Adjusted odds ratios for full masking, Stratified by Race/Ethnicity.

Variable	Black/African American OR (95% CI)	Hispanic OR (95% CI)	White OR (95% CI)	Other OR (95% CI)
Obs period				
1	Ref.	Ref.	Ref.	Ref.
2	1.45 (1.16, 1.81)	1.28 (0.97, 1.70)	1.99 (1.58, 2.52)	1.92 (1.18, 3.12)
3	1.36 (1.08, 1.71)	1.31 (0.96, 1.78)	1.75 (1.32, 2.24)	2.55 (1.47, 4.41)
Age Group				
Child	0.55 (0.42, 0.72)	0.80 (0.57, 1.12)	0.76 (0.52, 1.11)	1.35 (0.61, 2.99)
Teen	0.58 (0.39, 0.86)	0.67 (0.44, 1.01)	0.46 (0.32, 0.66)	0.86 (0.48, 1.56)
Adult	Ref.	Ref.	Ref.	Ref.
Senior	1.45 (1.08, 1.95)	1.66 (1.16, 2.35)	1.89 (1.50, 2.38)	2.26 (1.08, 4.72)
Signage				
No Signage	Ref.	Ref.	Ref.	Ref.
Mask-Negative Language	0.51 (0.33, 0.78)	0.56 (0.25, 1.27)	0.38 (0.17, 0.89)	0.71 (0.19, 2.62)
Mask-Positive Language	1.03 (0.79, 1.33)	1.05 (0.60, 1.84)	1.18 (0.76, 1.84)	1.03 (0.79, 1.33)
Distancing				
No	Ref.	Ref.	Ref.	Ref.
Yes	1.34 (1.10, 1.63)	1.17 (0.90, 1.52)	1.24 (1.02, 1.51)	0.85 (0.55, 1.33)
Location Type				
Walmart	Ref.	Ref.	Ref.	Ref.
Target	1.93 (1.24, 2.99)	1.13 (0.72, 1.77)	1.01 (0.74, 1.38)	1.31 (0.55, 3.14)
Publix grocery Shop	1.35 (0.97, 1.89)	1.44 (0.92, 2.25)	1.10 (0.85, 1.44)	1.57 (0.77, 3.19)
Campus Building	0.95 (0.56, 1.59)	0.73 (0.39, 1.39)	0.91 (0.61, 1.36)	0.56 (0.28, 1.11)
Mall	0.54 (0.36, 0.82)	0.73 (0.43, 1.25)	0.66 (0.44, 0.99)	0.69 (0.30, 1.58)
Weekday/Weekend				
Weekday	Ref.	Ref.	Ref.	Ref.
Weekend	0.97 (0.80, 1.17)	1.16 (0.91, 1.48)	1.41 (1.16, 1.71)	0.95 (0.62, 1.46)
Gender				
Male	Ref.	Ref.	Ref.	Ref.
Female	1.43 (1.19, 1.72)	1.51 (1.19, 1.91)	1.52 (1.26, 1.84)	1.74 (1.15, 2.64)

## Data Availability

The data presented in this study are available on request from the corresponding author.

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
