# Peer review of "Masking Behaviors in the Absence of Local Mandate—An Observational Study from Hillsborough County, Florida"

_ijerph, 2022, doi:10.3390/ijerph192315982_

Round 1

Reviewer 1 Report

Dear authors!

Thank you for the opportunity to review the manuscript!

Please find below some suggestions to improve the manuscript:

Introduction

Please add a reference for the claim In the face of crisis, people responded in a variety of ways, from adhering to the words of experts to rejecting all advice and public health guidance (line 34-35).

Methods

Please explain in more detail what it is Qualtrics™, and how the observation was madey.

Results

Please technically fix the table 1 (gender)

Text in line 157 please separate from the table 1.

This chapter is too extensive. It would be good to highlight the most important results, which are certainly not the representations in figures 1-5.

Discussion

Please, expand this chapter.

Emphasize in the limitations that you have approximately estimated the obtained age values race and ethnicity.

Author Response

Manuscript ID: ijerph-2018144

Title: Masking Behaviors in the Absence of Local Mandate – Observational Study from Hillsborough County, Florida

Authors: Jessica Pecoraro *, Chighaf Bakour, Alison Oberne, Amber Mehmood

We are very thankful to the reviewers for their constructive comments and suggestions. These have certainly guided us in revising the manuscript and producing a better and more refined submission. Reviewers’ comments are listed below, along with point-by-point responses and the corresponding changes to the manuscript. An updated version of the manuscript with tracked changes is attached, along with a clean version.

Sincerely,

Jessica Pecoraro, MPH, PMP, CPH

Introduction: Please add a reference for the claim In the face of crisis, people responded in a variety of ways, from adhering to the words of experts to rejecting all advice and public health guidance (line 34-35).

  • Response: The reference is added on line 46

Methods: Please explain in more detail what it is Qualtrics™, and how the observation was made.

  • Response: More details regarding Qualtrics and the observation and data collection procedures were added to the methods section (2.1: Population, setting, and data collection, line 112-119)

Results:

  1. Please technically fix the table 1 (gender).
    • Response: We fixed the table using sex in place of gender
  2. Text in line 157 please separate from the table 1.
    • Response: We added space between the text and table 1
  3. This chapter is too extensive. It would be good to highlight the most important results, which are certainly not the representations in figures 1-5.
    • Response: The number of figures was reduced from 5 to 3 to allow for less focus on descriptive results and more on the results of the regression analysis. Three remaining figures include Masking trends over time in children and adolescents/adults (line 203); Overall masking behaviors by race/ethnicity (line 218); and Masking behavior observed by local signage (line 232). The findings from the deleted figures (child-masking behavior by observation period and full masking trends over time by race/ ethnicity) are incorporated in text (lines 182-184 and 212-216)

Discussion

  1. Please, expand this chapter.
    • Response: Thank you for this suggestion. Each paragraph of this section was added to, expanding on the main themes discussed, including masking behaviors through the lens of health belief and risk perception. Implications for risk communication strategies for different target demographic groups was also added (lines 297-388) and the conclusion has been updated accordingly.
  2. Emphasize in the limitations that you have approximately estimated the obtained age values race and ethnicity.
    • Response: Thank you for this suggestion. Discussion surrounding this limitation was separated from the other mentioned limitations (it was previously part of a list contained in a single sentence). Discussing it separately allows emphasis to be brought to this specific limitation (lines 397-403).

Reviewer 2 Report

I have read the article entitled "Masking Behaviors in the Absence of Local Mandate – an Observational Study from Hillsborough County, Florida". The authors present interesting research. However, the work itself requires a bit of refinement to make it more readable. In the methods section, I suggest that you add information about the statistical program that was used to perform the calculations. You may also consider adding value labels to bar charts to make them easier to read. Especially in the case of low percentages, it is difficult to see the difference.

I would also suggest that the authors consider an additional limitation of the study. It is true that the research was carried out in different places and times, but it is likely that some of the respondents duplicated due to the selection of places of observation in the form of publicly available ones.

Author Response

Manuscript ID: ijerph-2018144

Title: Masking Behaviors in the Absence of Local Mandate – Observational Study from Hillsborough County, Florida

Authors: Jessica Pecoraro *, Chighaf Bakour, Alison Oberne, Amber Mehmood

We are very thankful to the reviewers for their constructive comments and suggestions. These have certainly guided us in revising the manuscript and producing a better and more refined submission. Reviewers’ comments are listed below, along with point-by-point responses and the corresponding changes to the manuscript. An updated version of the manuscript with tracked changes is attached, along with a clean version.

Sincerely,

Jessica Pecoraro, MPH, PMP, CPH

Reviewer 2

I have read the article entitled "Masking Behaviors in the Absence of Local Mandate – an Observational Study from Hillsborough County, Florida". The authors present interesting research. However, the work itself requires a bit of refinement to make it more readable. In the methods section, I suggest that you add information about the statistical program that was used to perform the calculations. You may also consider adding value labels to bar charts to make them easier to read. Especially in the case of low percentages, it is difficult to see the difference.

I would also suggest that the authors consider an additional limitation of the study. It is true that the research was carried out in different places and times, but it is likely that some of the respondents duplicated due to the selection of places of observation in the form of publicly available ones.

Response:

Thank you for your kind review and suggestions, we have revised the manuscript extensively to incorporate the following changes:

  • Information on the statistical program (SAS® 9.4) was added to the Statistical Analysis section (line 161)
  • Value labels were added to the charts (current figures 2, 3)
  • The suggested additional limitation of potential for duplicated observations was added to the discussion (line 397-399).